# Transcriptional analysis of wheat seedlings inoculated with *Fusarium culmorum* under continual exposure to disease defence inductors

**Zuzana Antalová**[1,2], **Dominik Bleša**[3,4], **Petr Martinek**[1], **Pavel Matušinsky** [ID][3,5]*

**1** Department of Plant Breeding and Genetics, Agrotest Fyto, Ltd, Kroměříž, Czech Republic, **2** Centre of the Region Haná for Biotechnological and Agricultural Research, Institute of Experimental Botany, Olomouc, Czech Republic, **3** Department of Plant Pathology, Agrotest Fyto, Ltd, Kroměříž, Czech Republic, **4** Department of Experimental Biology, Faculty of Science, Masaryk University, Brno, Czech Republic, **5** Department of Botany, Faculty of Science, Palacký University in Olomouc, Olomouc, Czech Republic

* matusinsky@vukrom.cz

**Data Availability Statement:** All relevant data are within the manuscript and its Supporting Information files.

## Abstract

A facultative parasite of cereals, *Fusarium culmorum* is a soil-, air- and seed-borne fungus causing foot and root rot, fusarium seedling blight, and especially Fusarium head blight, a spike disease leading to decreased yield and mycotoxin contamination of grain. In the present study, we tested changes in expression of wheat genes (*B2H2*, *ICS*, *PAL*, and *PR2*) involved in defence against diseases. We first compared expression of the analysed genes in seedlings of non-inoculated and artificially inoculated wheat (variety Bohemia). The second part of the experiment compared expression of these genes in seedlings grown under various treatment conditions. These treatments were chosen to determine the effects of prochloraz, sodium bicarbonate, ergosterol, aescin and potassium iodide on expression of the analysed defence genes. In addition to the inoculated and non-inoculated cultivar Bohemia, we additionally examined two other varieties of wheat with contrasting resistance to *Fusarium* sp. infection. These were the blue aleurone layer variety Scorpion that is susceptible to *Fusarium* sp. infection and variety V2-49-17 with yellow endosperm and partial resistance to *Fusarium* sp. infection. In this manner, we were able to compare potential effects of inductors upon defence gene expression among three varieties with different susceptibility to infection but also between inoculated and non-inoculated seedlings of a single variety. The lowest infection levels were detected in the sodium bicarbonate treatment. Sodium bicarbonate had not only negative influence on *Fusarium* growth but also positively affected expression of plant defence genes. Expression of the four marker genes shown to be important in plant defence was significantly affected by the treatments. The greatest upregulation in comparison to the water control was identified under all treatments for the *B2H2* gene. Only expression of *PAL* under the ergosterol and prochloraz treatments were not statistically significant.

**Funding:** This work and ZA, DB, PeM and PaM have been supported by the Ministry of Agriculture of the Czech Republic (project no. MZE RO1118). PaM have been supported by the Ministry of Agriculture of the Czech Republic (project no. QK1910197) and Internal Grant of Palacky University (project no. IGA_PrF_2020_003). PeM have been supported by the Ministry of Agriculture of the Czech Republic (project no. QK1910343). The funder Agrotest Fyto, Ltd provided support in the form of salaries for all authors, but did not have any additional role in the study design, data collection and analysis, decision to publish, or preparation of the manuscript. The specific roles of these authors are articulated in the 'author contributions' section.

**Competing interests:** All authors are emoloyed to Agrotest Fyto, Ltd This does not alter our adherence to all the PlosOne policies on sharing data and materials.

# Introduction

*Fusarium culmorum* is a ubiquitous soil-borne fungus with a highly competitive saprophytic capability. As a facultative parasite, it can cause foot and root rot [1]. *Fusarium culmorum* is also seed-borne and causes fusarium seedling blight when infected seed is used in sowing. Seedling blight can cause extensive damage to growing seedlings [2] that can lead to reduced plant establishment, number of heads per square meter, and also grain yield [3]. Fusarium head blight (FHB) is one of the most severe diseases responsible for decrease in grain yield and quality. Furthermore, presence of mycotoxins produced by this fungus (deoxynivalenol, niva-lenol, zearalenone, and many others) can harm human and animal health. FHB in wheat is mainly caused by *Fusarium graminearum*, *F. culmorum*, and *F. poae*. *Fusarium culmorum* infection is dominant in colder regions, such as north, east, and central Europe [1]. The major reservoirs of *Fusarium* sp. inoculum are crop residues on the soil surface. The fungus can survive on a wide range of living plant species (wheat, corn, barley, soybean, and rice) (see Bai and Shaner for a review [4]).

There are several means to fight this disease: use of fungicides, cultural practices, resistant cultivars, and biological agents [5]. Although seed treatment is used to control soil-borne infection caused by *Fusarium* spp. [6], there is no definite way to defeat this complex of Fusarium diseases. Efficacy of fungicide treatments against FHB is only 15–30% [7]. Fully resistant cultivars are not available to date, but some cultivars have useable levels of partial resistance that limit yield loss and mycotoxins accumulation [8]. FHB resistance has a quantitative nature and identification of responsible genes is difficult. Even though numerous quantitative trait loci have been described to date (see Duba et al. for a review [9]), just a few such genes have been definitively identified, sequenced, and their causal mutations determined. Kage et al. [10] identified an FHB resistance gene on chromosome 2DL as the *TaACT* gene encoding agmatine coumaroyl transferase. They suggest that several single nucleotide polymorphisms (SNPs) and two inversions may be important for gene function. The second identified gene, *Fhb1*, confers resistance in variety Sumai 3. It is pore-forming toxin-like (*PFT*) gene [11]. Further, a number of pathogenicity and virulence factors have been characterized [11, 12, 13].

The expression of defence-related genes also can be important in the plant's reaction against pathogens. *PR-1*, *PR-2* (glucanases), *PR-3* (chitinase), *PR-4* (thaumatin-like proteins), *PR-5*, and peroxidase have been shown to be induced in both resistant and susceptible cultivars after point inoculation [14]. These proteins were detected as early as 6 to 12 h after inoculation and peaked after 36 to 48 h. Earlier and greater expression of PR-4 and PR-5 transcripts were observed, however, in resistant cultivar Sumai 3 than in susceptible cultivar Wheaton [14]. Larger amounts of β-1,3-glucanase and chitinase enzymes also have been detected in resistant cultivar Sumai [15]. The overexpression of defence response genes in wheat could enhance FHB resistance in both greenhouse and field conditions.

A large number of organic and inorganic compounds have previously been described as affecting plant defence mechanisms [16, 17]. For example, such plant or fungal-derived compounds as monoterpenes or ergosterol can induce plant defence [18, 19].

Our study is focused upon comparing expression of different genes in non-inoculated and inoculated varieties and under various treatment conditions. The first part of the experiment compares expression of the genes *β-1,3-glucanase (PR2)*, *chitinase (B2H2)*, *phenylalanine ammonia-lyase (PAL)*, and *isochorismate synthase (ICS)* (the last two being genes from the salicylate pathway) in healthy seedlings versus seedlings of *Triticum aestivum* var. Bohemia artificially inoculated with *F. culmorum*. The second part of the experiment focused on how expression of these genes is influenced by different treatment solutions (based upon pro-chloraz, aescin, ergosterol, sodium bicarbonate and potassium iodide) within which seedlings

were grown. Prochloraz, potassium iodide and sodium bicarbonate were chosen for their anti-fungal effects in preliminary *in vitro* experiments (S1 Fig) and aescin and ergosterol because of their expected effect on plant triggered immunity. How these treatments influence expression of the aforementioned genes is compared with a water control in inoculated and healthy seedlings of Bohemia as well as in the moderately *Fusarium*-resistant yellow endosperm variety V2-49-17 and the susceptible blue aleurone layer variety Scorpion. The originality of this research consists in using artificially infected seeds during anthesis of mother plants and continuous exposure of seedlings to potential inductors.

## Materials and methods

### Plant material

Inoculated and non-inoculated (healthy) groups of wheat seeds (var. Bohemia) were used for the first part of the experiment. Inoculated and healthy seeds were acquired from plants grown under field conditions during the 2017 season. Inoculated seeds were collected from plants that had been sprayed during mid-anthesis phase by *F. culmorum* (tribe KM16902) macroconidia at concentration $5 \times 10^5$ conidia ml$^{-1}$. A previous study had shown a high level of virulence and strong production of deoxynivalenol (DON) by this tribe [20]. The seeds from inoculated and healthy variants were collected and stored at room temperature and low humidity. The plant material further consisted of two bread wheat cultivars differing in grain colour and in susceptibility to *F. culmorum* infection (V2-49-17 –yellow kernels, medium resistant to infection; Scorpion–blue aleurone layer, highly susceptible to infection).

### Growth chamber test under controlled conditions

Growth chamber test of 50 kernels of all three cultivars (Bohemia–inoculated and healthy; Scorpion, and V2-49-17) were laid with 1 cm separation distance into two layers of filtrate paper and rolled up. The rolls were immersed into the treatment solutions. Four replications (200 kernels in total) were made for each combination of cultivar and treatment solution. The treatment solutions consisted of 25 µg ml$^{-1}$ solution of ergosterol, 25 µg ml$^{-1}$ solution of aescin, 1 µg ml$^{-1}$ solution of prochloraz, 1% solution of potassium iodide and 0.1 M solution of sodium bicarbonate. The sodium bicarbonate solution was boiled at 120˚C for half an hour. During boiling, the sodium bicarbonate gradually decomposed to sodium carbonate, water and carbon dioxide. This reaction led to alkalization of pH. Distilled water was used as a control solution.

Seedlings were cultivated under controlled conditions (20˚C/18˚C, 12/12 h of light/dark) until the two-leaves growth stage. At this stage, the whole leaves were collected from three plants representing three biological replicates. The leaves showed no signs of *F. culmorum* infection. Symptoms of *F. culmorum* infection were visible only on the lower parts of plants and around the seeds (Fig 1). The leaves were immediately frozen in liquid nitrogen and preserved at −80˚C until RNA isolation. The numbers of infected seeds and seeds with no sign of infection were counted and the results were statistically analysed by ANOVA (Statistica 12 software).

### RNA isolation and qPCR

Leaves were homogenized in a TissueLyser II (Qiagen) for 2 minutes at 27 Hz. Caution was taken during homogenization to avoid sample melting. The homogenized samples were immediately placed into liquid nitrogen. The RNA was isolated using the RNeasy Plant Mini Kit

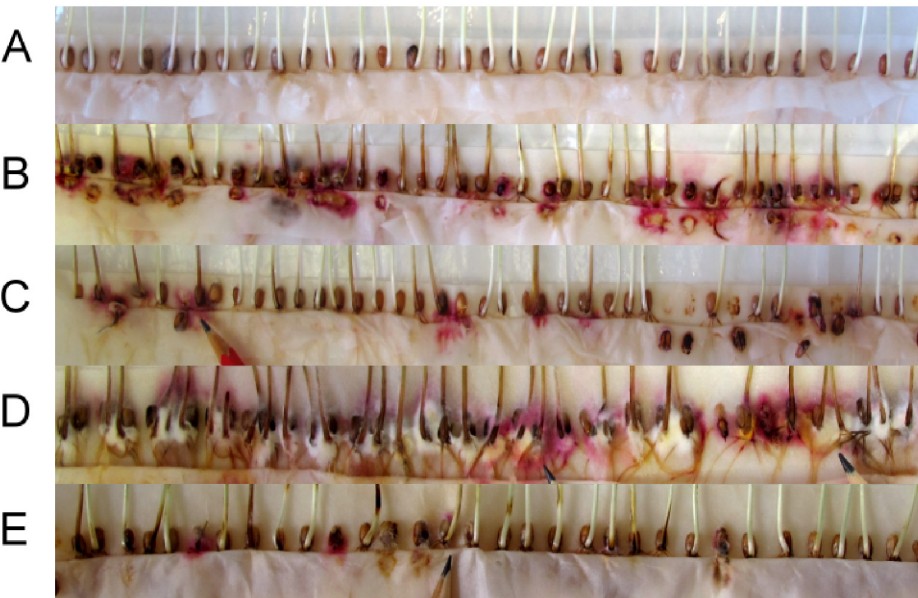

**Fig 1. Inoculated and non-inoculated seedlings of cv. Bohemia under different treatment conditions.** (A) non-inoculated seedlings in water, (B) inoculated seedlings in water, (C) inoculated seedlings in prochloraz, (D) inoculated seedlings in potassium iodide, (E) inoculated seedlings in sodium bicarbonate. Pink–white coloration around kernels and dark discoloration of coleoptiles and stem bases are symptoms of *F. culmorum* infection.

(Qiagen) while following the manufacturer's instructions. DNA was removed during the RNA purification using the RNase-Free DNase Set (Qiagen). The isolated RNA was stored at −80˚C. cDNA was synthesized using the Transcriptior High Fidelity cDNA Synthesis Kit (Roche Diagnostics, Mannheim, Germany) according to the manufacturer's instructions with 1 μg of total RNA and anchored-oligo (dT) primers. The concentration of cDNA was measured with Qubit (Thermo Fisher Scientific) and cDNA was diluted to concentration 10 ng μl$^{-1}$. The expression analysis of the chosen plant defence genes (chitinase (*B2H2*), β-1,3-glucanase (*PR2*), isochorismate synthase (*ICS*), and phenylalanine ammonia-lyase (PAL)) was performed using the CFX96TM Real-Time PCR Detection System (Bio-Rad). The qPCR mix consisted of 1× SYBR Green (Top Bio), 0.2 μM forward and reverse primers (Table 1), 15 ng cDNA, and water to final volume 15 μl. The reference gene was glyceraldehyde-3-phosphate dehydrogenase (GAPDH) according to Travella et al. (2006) [21] and Sun et al. (2014) [22]. The control sample consisted of equal amounts of cDNA from all three replicates of healthy, untreated Bohemia seedlings diluted to 10 ng μl$^{-1}$. The primers specificity and presence of primer dimers were verified by melting analysis. The data were analysed using the $2^{-\Delta\Delta Cq}$ method with CFX Manager 3.0 software (Bio-Rad, USA). Three biological as well as three technical replicates were run.

**Table 1. Primer pairs used in the study.** Names, sequences of forward and reverse primers, publication sources of primer pairs, and gene functions are listed.

| Gene name | Forward primer | Reverse primer | Publication | Function |
|---|---|---|---|---|
| *B2H2* | TCTATCGAAACGCCATTGTTACA | AGAGGCCGTTCGCATAGTCA | [42] | chitinase |
| *PR2* | CCGCACAAGACACCTCAAGATA | CGATGCCCTTGGTTTGGTAGA | [43] | β-1,3-glucanase |
| *PAL* | TTGATGAAGCCGAAGCAGGACC | ATGGGGGTGCCTTGGAAGTTGC | [44] | salicylate pathway |
| *ICS* | AGAAATGAGGACGACGAGTTTGAC | CCAAGTAGTGCTGATCTAATCCCAA | [44] | salicylate pathway |
| *GAPDH* | TTAGACTTGCGAAGCCAGCA | AAATGCCCTTGAGGTTTCCC | [22] | reference gene |

## Results

### Determination of *Fusarium* infection level in growth chamber test

Inoculated and healthy seeds of wheat cv. Bohemia were treated with different solutions: water, prochloraz, aescin, ergosterol, sodium bicarbonate and potassium iodide. Inoculated seeds contained high levels of *F. culmorum* DNA (analysed by qPCR, S1 File), the mean of three replications being 5,048 µg kg$^{-1}$ of DON (analysed by ELISA, S1 File). No *F. culmorum* DNA was detected in the non-inoculated seeds, and their DON content (if any) was under the detection limit. The plants were visually inspected at the two-leaves stage for presence of *Fusarium* infection. The number of infected seeds under every treatment was compared to that for the control (treated only with water). The presence of *Fusarium* infection was detectable by pink–white mycelia growing around kernels and dark discoloration of the coleoptile and stem (Fig 1).

In the variant without inoculation there were no infected seeds. On the contrary, the inoculated seeds that had been submerged in water showed high level of infection (Fig 1). This level of infection was decreased under every treatment except for that of potassium iodide. The level of infection in the potassium iodide treatment group was even increased in comparison to the water-treated inoculated seeds. In evaluating the 200 seeds from each combination, significant differences were detected between individual groups. The results suggest that the lowest level of infection in inoculated seeds was detected in the sodium bicarbonate treatment, followed (in order from lowest to highest) by prochloraz, ergosterol, aescin, water, and potassium iodide (Fig 2). Thus, the treatment with 0.1M sodium bicarbonate was more potent in suppressing fungal growth than was the treatment with 1 µg ml$^{-1}$ prochloraz.

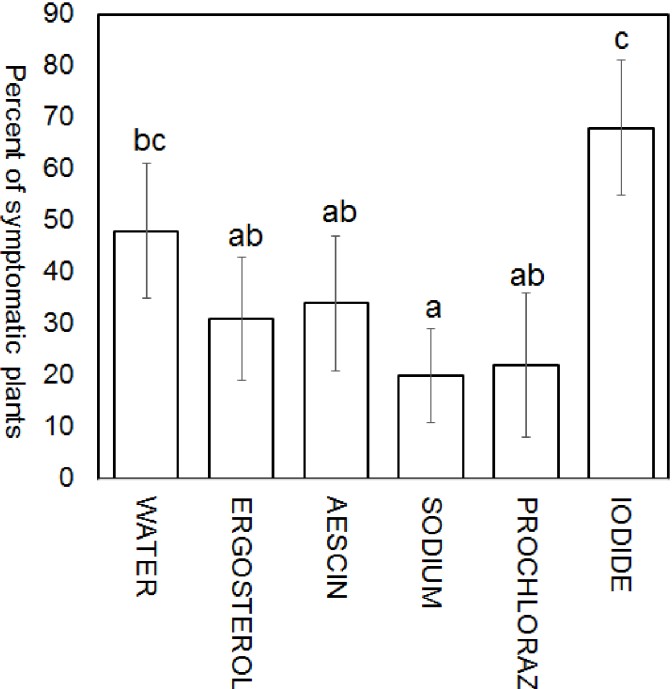

**Fig 2. Percentage of *F. culmorum*-infected plants in different treatments with cultivar Bohemia.** Two hundred inoculated seeds were evaluated for each treatment solution (water [control], ergosterol, aescin, sodium bicarbonate, prochloraz and potassium iodide). Error bars indicate 95% confidence intervals around the mean.

## Expression levels of four genes involved in plant–pathogen interaction in wheat seedlings

The expression of four important plant defence genes (*B2H2*, *ICS*, *PAL*, and *PR2*) against *Fusarium* infection was detected. These genes were analysed in three technical and biological replications of each studied variety under each treatment. The fold differences (FDs) in their expression levels were first compared among the reference (water control) and other experimental conditions (ergosterol, aescin, sodium bicarbonate, prochloraz, and potassium iodide) of all studied varieties (Table 2). In this manner, the efficiency of different treatments in enhancing expression of plant defence genes with and without pathogen infection was examined. The treatments did not influence expression of the genes in an even manner, as expression was stronger in some genes and weaker in others. Only insignificant changes were detected in inoculated Bohemia under the ergosterol treatment (*ICS*, *PAL*) as well as in Scorpion under the aescin treatment (*PAL*).

*B2H2* expression was upregulated by almost all treatments in all studied varieties (Table 2). The highest expression level of this gene was achieved under the sodium bicarbonate (V2-49-17) treatment followed by the prochloraz (V2-49-17) and potassium iodide (healthy Bohemia) treatments. It was downregulated just under the ergosterol (healthy Bohemia and Scorpion), aescin (Scorpion), and prochloraz (Scorpion) treatments. This gene achieved the largest FD (up to 7,370; potassium iodide) in comparison to the other genes. The FD was lowest in inoculated Bohemia. The FD of *B2H2* was in some cases near to 1,000 in comparison to the water control. This high FD was not achieved solely in the inoculated Bohemia and the Scorpion variety. In inoculated Bohemia, for example, the FD ranged from 0.95 under the sodium to 7.45 under the iodide treatment. In healthy Bohemia, the expression ranged from −0.99 to 7,370 FD in comparison to the water control. A similarly large increase of expression in

**Table 2. Expression levels of four chosen genes in leaves of healthy and inoculated plants of wheat variety Bohemia and healthy V2-49-17 and Scorpion varieties.**
The expression of four genes (*B2H2*, *ICS*, *PAL*, and *PR2*) were detected in healthy and inoculated Bohemia, V2-49-17 variety with yellow endosperm, and cv. Scorpion with blue aleurone layer in different treatment conditions (ergosterol, aescin, sodium bicarbonate, prochloraz and potassium iodide) and indicated as a fold difference (FD) to control (treatment water).

| Variant | Gene | Ergosterol | | Aescin | | Sodium bicarbonate | | Prochloraz | | Potassium iodide | |
|---|---|---|---|---|---|---|---|---|---|---|---|
| Healthy Bohemia | *B2H2* | -0,99 | *** | 10,16 | **** | 838,05 | **** | 72,96 | **** | 7370,13 | **** |
| | *ICS* | -0,76 | *** | 3,76 | **** | 2,71 | *** | 4,53 | **** | -0,39 | *** |
| | *PAL* | -0,83 | ** | 0,94 | ** | 5,99 | **** | 5,06 | **** | 4,98 | **** |
| | *PR2* | -0,84 | ** | 3,97 | **** | 1,07 | ** | 0,42 | ** | 20,75 | **** |
| Inoculated Bohemia | *B2H2* | 2,29 | ** | 2,80 | ** | 0,95 | ** | 5,20 | **** | 7,45 | **** |
| | *ICS* | -0,27 | | 1,33 | *** | 0,41 | ** | -0,85 | *** | -0,57 | ** |
| | *PAL* | 2,10 | ** | 1,98 | ** | 4,99 | **** | 7,12 | **** | 3,86 | **** |
| | *PR2* | 1,30 | *** | 3,11 | **** | 1,96 | *** | 1,31 | *** | 30,80 | **** |
| V2-49-17 variety with yellow endosperm | *B2H2* | 42,73 | **** | 46,67 | **** | 1060,53 | **** | 2040,09 | **** | 1362,64 | **** |
| | *ICS* | 1,21 | **** | 8,27 | **** | 6,26 | **** | 12,29 | **** | -0,33 | *** |
| | *PAL* | 8,57 | *** | 4,39 | *** | 57,79 | **** | 82,05 | **** | -1,00 | * |
| | *PR2* | 23,42 | **** | 58,49 | **** | 25,16 | **** | 20,09 | **** | 48,52 | **** |
| Scorpion with blue aleurone layer | *B2H2* | -0,98 | **** | -0,98 | **** | 122,92 | **** | -0,42 | *** | 44,48 | **** |
| | *ICS* | 1,22 | **** | -0,69 | *** | -0,87 | *** | 1,59 | **** | -0,09 | * |
| | *PAL* | 5,85 | *** | -0,02 | | 14,02 | **** | 43,83 | **** | -1,00 | *** |
| | *PR2* | 4,08 | **** | 2,73 | **** | 20,16 | **** | 21,71 | **** | 49,85 | **** |

Significant fold differences between water control and experimental treatments are indicated by asterisks (P < 0.05 (*); P < 0.01 (**); P < 0.001 (***); P < 0.0001 (****)). The statistical analyses were carried out separately within each gene and treatment.

comparison to the water control as in healthy Bohemia was identified also in V2-49-17. The V2-49-17 variety showed significant increase of *B2H2* expression in comparison to healthy Bohemia while the expression of all other genes was lower.

The *ICS* gene manifested the smallest fold increase under almost all analysed treatments. Its expression was often downregulated under some experimental treatments in some analysed varieties (Table 2). The largest fold difference was detected in the V2-49-17 variety under the prochloraz treatment, the smallest in inoculated Bohemia under the prochloraz treatment. Under the iodide treatment, all FDs were in negative values for all analysed varieties.

FDs for *PAL* expression were increased under almost every treatment. These increases were not to such large extent as seen for *B2H2*. The largest FDs were achieved under prochloraz (82.05 FD) and sodium bicarbonate (57.79 FD) treatments in the V2-49-17 variety. Downregulation of *PAL* expression in comparison to the water control was identified under the ergosterol treatment in healthy Bohemia, under potassium iodide treatment in V2-49-17, and under potassium iodide and aescin treatments in the Scorpion variety (Table 2). The lowest FD was detected under iodide in the yellow variety.

The FDs for *PR2* expression in comparison to the water control were elevated under almost all treatments and in all analysed varieties except for the ergosterol treatment in healthy Bohemia. The largest FD was seen in the V2-49-17 variety under all treatments except for prochloraz and iodide, in which cases the FDs were greater in the Scorpion variety. The largest FD for *PR2* was observed under the iodide treatment (from 20.75 to 49.85 FD). The lowest FD in almost all cases was found in healthy Bohemia (Table 2).

We further compared the expression of all four genes between the inoculated and healthy Bohemia under all treatments. The strongest *B2H2* expression in inoculated Bohemia was identified under potassium iodide treatment and the weakest under the control. The strongest *B2H2* expression in healthy Bohemia was identified under potassium iodide treatment followed by that for sodium bicarbonate (Fig 3). The expression of *B2H2* was increased in inoculated Bohemia under all treatments.

Expression of *ICS* was significantly downregulated in inoculated Bohemia under almost all treatments, the exception being the ergosterol treatment, in which case the difference was not statistically significant (Fig 3). The strongest *ICS* expression was detected under the prochloraz treatment in healthy Bohemia. The weakest was under the iodide and ergosterol treatments.

Comparison of healthy versus inoculated Bohemia showed increase of *PAL* expression in inoculated Bohemia under all treatments. The strongest expression of *PAL* in healthy Bohemia was identified under the sodium bicarbonate treatment. The greatest expression in inoculated Bohemia was identified under the prochloraz and sodium bicarbonate treatments (Fig 3).

*PR2* expression was elevated in all treatments other than the aescin treatment in inoculated Bohemia. The difference between healthy and inoculated Bohemia under the aescin treatment was not statistically significant. The expression of *PR2* in inoculated Bohemia was elevated under the iodide (in comparison to healthy Bohemia), and ergosterol treatments with high significance. Small increases were detected under the water and sodium bicarbonate treatments (Fig 3).

## Discussion

In current study was tested the expression of chosen marker genes of wheat seedlings after various treatments by potential plant defence inductors. The effect of plant defence inductors was previously widely studied [18, 19] and their effect on defence genes expression was taken to the account. Effect of chitinase genes in increasing plant resistance to fungal diseases has been observed in previous studies (see Fahmy et al. for a review [23]). Transgenic wheat with barley chitinase II was shown to be resistant against powdery mildew, leaf rust pathogens,

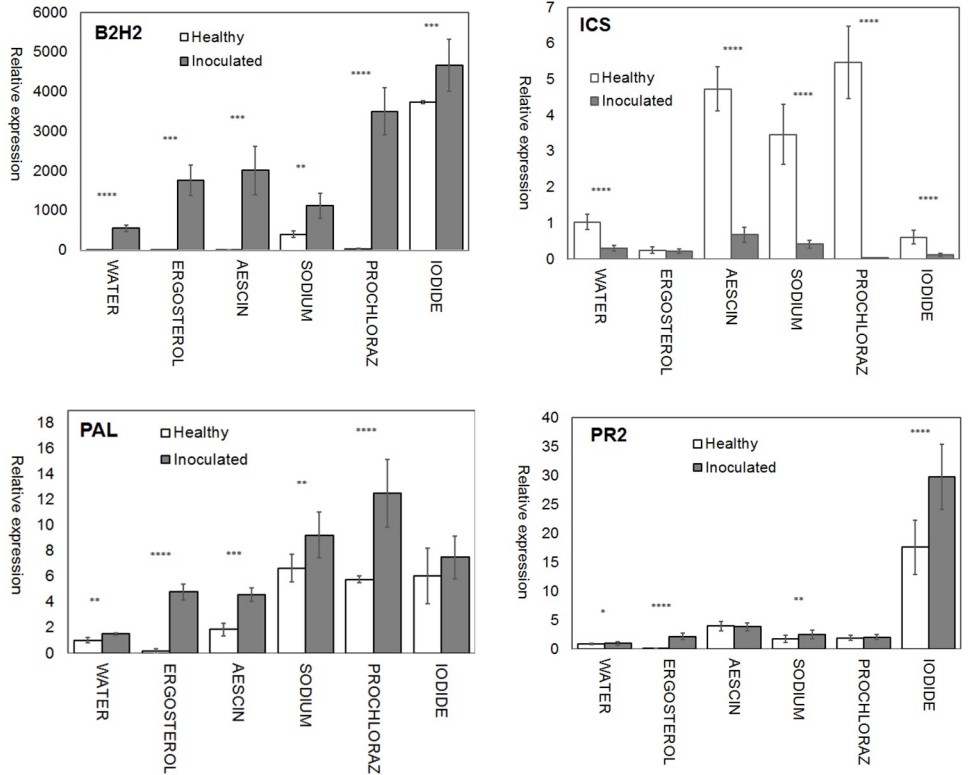

**Fig 3. Expression profiles of *B2H2*, *ICS*, *PAL*, and *PR2* in leaves of inoculated and healthy plants of wheat variety Bohemia under different treatment conditions (A, B, C, D).** Expression levels were relative to healthy cv. Bohemia seeds and were normalized with the wheat reference gene *GAPDH*. Expression levels shown are mean values and standard deviation for three replications. Statistically significant differences between healthy and inoculated cv. Bohemia plants are indicated by asterisks above every treatment ($P < 0.05$ (*); $P < 0.01$ (**); $P < 0.001$ (***); $P < 0.0001$ (****).

and *F. graminearum* [24, 25, 26]. Chitinase from wheat, barley, and maize kernels has been shown to inhibit hyphal elongation of the fungi [27]. In the present study, expression of the chitinase gene in the V2-49-17 variety was greater than was its expression in healthy Bohemia under the control variant (water) as well as under the experimental treatments. This higher value of chitinase expression can be connected to partial resistance of the yellow variety to *Fusarium* sp. infection. Higher chitinase levels in resistant cultivars already have been detected in previous studies [28, 29]. The largest fold increase for *B2H2* expression in healthy Bohemia was detected under the potassium iodide treatment. This 7,000 FD could be explained by the negative effect of this treatment on seedlings growth. This retardation of growth was connected to increase of *F. culmorum* infection in potassium iodide-treated varieties regardless of variety or presence of *Fusarium* infection. However, more than 1,000 FD [1,363] in comparison to the water control was detected just in V2-49-17. The growth retardation of seedlings under iodide treatment in our experiment is substantiated by the previous study of Brenchley [30], who reported a negative effect of high concentration of potassium iodide on germination of barley seeds and even of low concentration on the survival of barley seedlings. Iodine at concentration 10 ppm was observed to be toxic to barley. Nevertheless, a concentration 0.5–1 ppm had a positive effect on barley plants [31]. The iodine is most toxic in its iodide form [32]. This is the form we used in our experiments at 10,000 ppm concentration. Used concentration inhibited *F. culmorum* growth in preliminary

*in vitro* test (S1 Fig). The observed toxicity of potassium iodide for wheat seedlings is thus understandable and such a high concentration suppress natural plant immunity, thus allowing us to see the effect of immunity on the intensity of the pathogen development (Fig 1). The strong expression of *B2H2* is just a result of this treatment and does not reflect an effect of potassium iodide's increasing resistance in the plant. This inhibitory effect was more potent than the effects of sodium bicarbonate and prochloraz. Significant increase of *B2H2* expression could be seen also in the sodium bicarbonate treatment. This increase exceeded those under the prochloraz treatments. It can be seen across all analysed varieties with the exception of the inoculated Bohemia. Our results imply that sodium bicarbonate is significantly effective in enhancing expression of defence genes. Contrary to potassium iodide, sodium bicarbonate has no negative effect on wheat seedlings' growth. Comparison of all treatments in inoculated wheat showed that the lowest numbers of seedlings with detectable *Fusarium* infection at the three-leaves stage were under the prochloraz and sodium bicarbonate treatments. This suggests that sodium bicarbonate has potential for increasing plant resistance. The addition of sodium bicarbonate to experimental treatments was conditioned by its alkaline pH. This was in accordance with studies showing inhibition of *TRI* genes expression in *Fusarium* by alkaline pH [33, 34, 35]. *TRI* genes expression is important for synthesis of DON, which is known to be a virulent factor aiding in the establishment and propagation of *Fusarium* infection within the spikes [36, 37]. In preliminary experiments, the sodium bicarbonate showed potential for inhibiting *Fusarium* growth *in vitro* (S1 Fig). Indeed, the sodium bicarbonate showed great potential also *in planta* (Fig 1). Sodium bicarbonate had not only a negative influence on *Fusarium* growth but also a positive effect on expression of plant defence genes. The sodium bicarbonate has been shown to be potent in inhibiting growth also of other *Fusarium* species [38, 39].

We analysed the SA pathway's function in plant defence by examining expression of the two genes *ICS* and *PAL*, both of whose biosynthetic pathways are known to be involved in SA production within *Arabidopsis* [40]. Hao et al. [40] had previously detected that suppression of the *ICS* gene compromised plant resistance to *F. graminearum* but that similar suppression of *PAL* genes had no significant effect. Those authors also found that *F. graminearum*-inoculated plants with stronger expression of *ICS* were comparable to wild-type control plants [40] and that plants with *ICS* suppressed did not accumulate SA during pathogen infection and were more susceptible to *Fusarium*. In the present study, the suppression of growth and higher rate of *Fusarium* infection connected to lower *ICS* expression were detectable predominantly under the potassium iodide treatment, where *ICS* had negative FD in comparison to the water treatment in every analysed variety. Similarly, *ICS* was downregulated under all treatments in the inoculated wheat cv. Bohemia. On the other hand, *ICS* expression was upregulated in all other treatments except for a few exceptions in Scorpion and inoculated Bohemia. It was upregulated in V2-49-17 under all treatments other than that of potassium iodide, which can be connected to this variety's partial *Fusarium* resistance. Upregulation of *ICS* in inoculated Bohemia in comparison to the water control was detected only under sodium bicarbonate and aescin treatments, which can indicate effects of these treatments on expression of defence genes. Hao et al. [40] have suggested that *ICS* plays a unique role in SA biosynthesis in barley, which, in turn, confers a basal resistance to *F. graminearum* by modulating the accumulation of $H_2O_2$, $O_2$, and reactive oxygen-associated enzymatic activities. In the present study, the greatest increase in *PAL* expression was detected in the V2-49-17 and Scorpion varieties. We found no correlation between higher *PAL* expression and the level of resistance to *F. culmorum*, because V2-49-17 is partially resistant but Scorpion is susceptible. In healthy versus inoculated Bohemia, *PAL* expression was generally increased under most treatments and especially under the potassium iodide treatments. Wildermuth et al. had detected increased *PAL*

and decreased *ICS* expression after *F. graminearum* infection [41]. They also found a difference in timing whereby earlier increase of *PAL* expression was detected in a partially resistant variety (Wangshuibai). They found no difference, however, between resistant and susceptible varieties in the timing of decrease in *ICS* expression.

## Conclusion

We conclude that prochloraz and sodium bicarbonate has the greatest potential for suppression of fungal development without having a negative effect on plant growth. According to our findings, the sodium bicarbonate had not only a negative influence on *Fusarium* growth but also a positive effect on upregulating the expression of plant defence genes.

## Supporting information

**S1 File. Confirmation of *F. culmorum* presence in inoculated seeds.**
(DOCX)

**S1 Fig. Inhibitory effect of different treatments against F. culmorum growth on PDA (potato dextrose agar) on dark under 16˚C for 5 days (Petri dishes diameter 90 mm).** HCl pH4 –PDA balanced to pH4 by HCl, NaOH pH10 –PDA balanced to pH10 by NaOH, IODIDE–PDA with potassium iodide (1% concentration), SODIUM–PDA with sodium bicarbonate (0.1 M), ACETATE–PDA with ammonium acetate (0.1 M), SDHI–PDA with fluxapyroxad (1 µg ml$^{-1}$), QoI–PDA with picoxystrobin (1 µg ml$^{-1}$), DMI–PDA with prochloraz (1 µg ml$^{-1}$).
(EPS)

## Acknowledgments

We thank Peter Antal for his advising us with the establishment of *in vitro* experiments.

## Author Contributions

**Conceptualization:** Zuzana Antalová, Petr Martinek, Pavel Matušinsky.

**Data curation:** Zuzana Antalová, Dominik Bleša.

**Formal analysis:** Zuzana Antalová, Petr Martinek.

**Funding acquisition:** Petr Martinek, Pavel Matušinsky.

**Investigation:** Zuzana Antalová, Dominik Bleša.

**Methodology:** Zuzana Antalová, Dominik Bleša.

**Project administration:** Petr Martinek.

**Resources:** Petr Martinek.

**Software:** Zuzana Antalová, Dominik Bleša.

**Supervision:** Petr Martinek, Pavel Matušinsky.

**Validation:** Petr Martinek.

**Writing – original draft:** Zuzana Antalová, Dominik Bleša.

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
