## [Decision Letter · Decision Letter 0]

4 Nov 2019

PONE-D-19-28330

Transcriptional analysis of wheat seedlings inoculated with Fusarium culmorum under continual exposure to disease defence inductors

PLOS ONE

Dear Dr. Matusinsky,

Thank you for submitting your manuscript to PLOS ONE. After careful consideration, we feel that it has merit but does not fully meet PLOS ONE’s publication criteria as it currently stands. Therefore, we invite you to submit a revised version of the manuscript that addresses the points raised during the review process.

We would appreciate receiving your revised manuscript by Dec 19 2019 11:59PM. To enhance the reproducibility of your results, we recommend that if applicable you deposit your laboratory protocols in protocols.io, where a protocol can be assigned its own identifier (DOI) such that it can be cited independently in the future. For instructions see: http://journals.plos.org/plosone/s/submission-guidelines#loc-laboratory-protocols

We look forward to receiving your revised manuscript.

Kind regards,

Binod Bihari Sahu, Ph.D.

Academic Editor

PLOS ONE

Journal Requirements:

'The authors have declared that no competing interests exist.'

We note that one or more of the authors are employed by a commercial company:  Agrotest Fyto, Ltd.

Additional Editor Comments (if provided):

Please comply the concerns raised by the reviewer before it is ready for publication.

Reviewers' comments:

Reviewer's Responses to Questions

**Comments to the Author**

1. Is the manuscript technically sound, and do the data support the conclusions?

Reviewer #1: Yes

2. Has the statistical analysis been performed appropriately and rigorously? 

Reviewer #1: Yes

3. Have the authors made all data underlying the findings in their manuscript fully available?

Reviewer #1: No

4. Is the manuscript presented in an intelligible fashion and written in standard English?

Reviewer #1: Yes

5. Review Comments to the Author

Reviewer #1: This is an interesting paper that shows the effect on Fusarium inoculation in susceptible and resistant wheat of several different treatments in an attempt to identify potential treatments that can inhibit Fusarium inoculation and DON production during wheat production. I di have several concerns about the manuscript that are discussed below.

There is little to no indication why the authors choose the specific treatments of prochloraz, sodium bicarbonate, ergosterol, aescin and potassium iodide for this research. The authors need to provide an explanation of why these specific treatments were chosen.

The heat map of expression levels in Figure 3 is frankly quite confusing. How are shades of light to dark chosen; Is there an expression range for each shade of grey to black? How do you distinguish between up and down regulation relative to the treatment? Both the figure explanation and perhaps the figure itself needs some modification to make the heat map data/results clearer to the reader.

I think the realtime QPCR and ELIZA results should be included as supplementary data.

In the results and discussion sections, it is clear that the 10,000 ppm iodide treatment had a deleterious effect on germination and seedling growth. The literature cited (31-33) document this growth inhibition. Consequently, it is not clear why the 10,000 ppm concentration was chosen. Is this the only concentration that inhibits Fusarium growth? I am not sure of the value of having potassium-iodine included as a treatment in this paper unless it is actively thought of as a potential treatment to wheat to inhibit Fusarium growth and DON production during wheat production.

The authors, in the conclusion, state that sodium bicarbonate is a possible treatment of wheat to reduce infection and growth of Fusarium. I agree with this conclusion. However, I see that prochloraz shows results very similar to sodium bicarbonate yet it is not identified as a possible treatment. I do not understand why both of these treatments are not indicated as such in the conclusion unless prochloraz has a deleterious effect on plant growth. If this is the case there is no data presented that suggests that.

6. PLOS authors have the option to publish the peer review history of their article (what does this mean?). If published, this will include your full peer review and any attached files.

Reviewer #1: No

---

## [Author Response · Author response to Decision Letter 0]

29 Nov 2019

Dear Editor and Reviewers

Thank you for your comments and for reviewing our article. Please find below our reaction to all recommendations. 

1. Please ensure that your manuscript meets PLOS ONE's style requirements, including those for file naming. The PLOS ONE style templates can be found at http://www.journals.plos.org/plosone/s/file?id =wjVg/PLOSOne_formatting_sample_main_body.pdf and http://www.journals.plos.org/plosone/s/file?id=ba62/PLOSOne_formatting_sample_title_authors_affiliations.pdf

We checked PlosOne style requirements according to style formats. We made these changes:

We formatted title and subtitle. We removed ZIP and Postal Codes and street addresses from affiliations. We included departments (Department of plant breeding and genetics and Department of plant pathology) in affiliation of Agrotest. We improved email address format of corresponding author. We set the double-spacing format in whole manuscript. We used Level 1 and 2 heading for sections and subsections. We changed citation of references to square brackets. We removed funding from Acknowledgments. We included “doi” codes to the list of references.

We included data in Supporting Information files (S1 Supporting Information and S1 Figure) where necessary we removed the mentioned phrases.

'The authors have declared that no competing interests exist.'

We note that one or more of the authors are employed by a commercial company: Agrotest Fyto, Ltd.

We stated in the Competing Interest section that: 

“The authors have declared that no competing interests exist. There are no patents, products in development or marked product to declare. This does not alter our adherence to all the PlosOne policies on sharing data and materials.”

Our commercial affiliation did not play a role in our study. We included the following statement within our amended Funding Statement.

“This work and ZA, DB, PeM and PaM have been supported by the Ministry of Agriculture of the Czech Republic (project no. MZE RO1118). PaM have been supported by the Ministry of Agriculture of the Czech Republic (project no. QK1910197) and Internal Grant of Palacky University (project no. IGA_PrF_2019_004). PeM have been supported by the Ministry of Agriculture of the Czech Republic (project no. QK1910343). The funder Agrotest Fyto, Ltd provided support in the form of salaries for all authors, but did not have any additional role in the study design, data collection and analysis, decision to publish, or preparation of the manuscript. The specific roles of these authors are articulated in the ‘author contributions’ section.”

5. Please also provide an updated Competing Interests Statement declaring this commercial affiliation along with any other relevant declarations relating to employment, consultancy, patents, products in development, or marketed products, etc. 

We stated in the Competing Interest section that: 

“The authors have declared that no competing interests exist. There are no patents, products in development or marked product to declare. This does not alter our adherence to all the PlosOne policies on sharing data and materials.”

We checked all figures by PACE and converted them to right format.

 

Response to Reviewers

1. Reviewer #1: This is an interesting paper that shows the effect on Fusarium inoculation in susceptible and resistant wheat of several different treatments in an attempt to identify potential treatments that can inhibit Fusarium inoculation and DON production during wheat production. I did have several concerns about the manuscript that are discussed below.

We thank reviewers for helpful comments, critical reading and the interest in improving the manuscript quality. We have processed the comments on our best. We hope that our amendments to the manuscript are suitable. All changes have been tracked using the revision mode in Word.

2. There is little to no indication why the authors choose the specific treatments of prochloraz, sodium bicarbonate, ergosterol, aescin and potassium iodide for this research. The authors need to provide an explanation of why these specific treatments were chosen.

We agree that this point was not clear enough. We have added appropriate explanation to the manuscript in Introduction part and also in Supplementary Information (S1 Figure):

“Prochloraz, potassium iodide and sodium bicarbonate were chosen for their antifungal effects in preliminary in vitro experiments (S1 Figure) and aescin and ergosterol because of their expected effect on plant triggered immunity.”

3. The heat map of expression levels in Figure 3 is frankly quite confusing. How are shades of light to dark chosen; Is there an expression range for each shade of grey to black? How do you distinguish between up and down regulation relative to the treatment? Both the figure explanation and perhaps the figure itself needs some modification to make the heat map data/results clearer to the reader.

We accepted the reviewer opinion and for a better clarity, the heat map (Figure 3) has been rearranged to the table (Table 2). Depending on these changes, we renumbered the other following figures. 

4. I think the realtime QPCR and ELIZA results should be included as supplementary data.

We accepted the reviewer opinion and we moved qPCR and ELISA to the Supplementary Information section (Supplementary Information 1).

5. In the results and discussion sections, it is clear that the 10,000 ppm iodide treatment had a deleterious effect on germination and seedling growth. The literature cited (31-33) document this growth inhibition. Consequently, it is not clear why the 10,000 ppm concentration was chosen. Is this the only concentration that inhibits Fusarium growth? I am not sure of the value of having potassium-iodine included as a treatment in this paper unless it is actively thought of as a potential treatment to wheat to inhibit Fusarium growth and DON production during wheat production.

We thank the reviewer for comment regarding iodide treatment. We added explanation to the manuscript and appropriate results of in vitro preliminary test to the Supplementary Information (S1 Figure).

„Used concentration inhibited F. culmorum growth in preliminary in vitro test (S1 Figure). The observed toxicity of potassium iodide for wheat seedlings is thus understandable and such a high concentration suppress natural plant immunity, thus allowing us to see the effect of immunity on the intensity of the pathogen development (Fig 1).“

6. The authors, in the conclusion, state that sodium bicarbonate is a possible treatment of wheat to reduce infection and growth of Fusarium. I agree with this conclusion. However, I see that prochloraz shows results very similar to sodium bicarbonate yet it is not identified as a possible treatment. I do not understand why both of these treatments are not indicated as such in the conclusion unless prochloraz has a deleterious effect on plant growth. If this is the case there is no data presented that suggests that.

Indeed, the prochloraz is not in conclusions so it was corrected in the manuscript and we changed the first sentence in Conclusion to this form:

“We conclude that prochloraz and sodium bicarbonate has the greatest potential for suppression of fungal development without having a negative effect on plant growth.”

---

## [Editor Report · Decision Letter 1]

21 Jan 2020

Transcriptional analysis of wheat seedlings inoculated with Fusarium culmorum under continual exposure to disease defence inductors

PONE-D-19-28330R1

Dear Dr. Matusinsky,

We are pleased to inform you that your manuscript has been judged scientifically suitable for publication and will be formally accepted for publication once it complies with all outstanding technical requirements.

With kind regards,

Binod Bihari Sahu, Ph.D.

Academic Editor

PLOS ONE
---

## [Editor Report · Acceptance letter]

24 Jan 2020

PONE-D-19-28330R1 

Transcriptional analysis of wheat seedlings inoculated with *Fusarium culmorum* under continual exposure to disease defence inductors 

Dear Dr. Matušinsky:

I am pleased to inform you that your manuscript has been deemed suitable for publication in PLOS ONE. Congratulations! Your manuscript is now with our production department. 

With kind regards,

on behalf of

Dr. Binod Bihari Sahu 

Academic Editor

PLOS ONE